# MedMeta: A Benchmark for LLMs in Synthesizing Meta-Analysis Conclusion

## Abstract

Large language models (LLMs) have saturated standard medical benchmarks, yet their ability to synthesize conclusions from multiple sources remains critically underexplored. To address this gap, we introduce MedMeta, the first benchmark for evaluating conclusion synthesis from medical meta-analyses. MedMeta comprises 81 meta-analyses and evaluates models under both Retrieval-Augmented Generation (RAG) and parametric-only workflows. Our findings underscore the critical importance of information grounding: RAG consistently and significantly outperforms Parametric-CoT across models. In contrast, the benefits of domain-specific fine-tuning are marginal and largely neutralized when external material is provided. We also uncover a critical, universal vulnerability: all tested models fail to identify and reject factually incorrect evidence, instead synthesizing it into coherent but false conclusions. Notably, even under ideal RAG conditions with oracle retrieval, the performance of current LLMs remains moderate, with the top-performing model scoring 3.17 out of 5.0. Our evaluation is grounded in an LLM-as-a-judge protocol. We validate this approach against human medical experts, showing a high Pearson's r (0.81) and negligible systematic bias in Bland–Altman analysis, establishing it as a reliable proxy for experts and a scalable assessment method. MedMeta establishes a challenging new benchmark and demonstrates that developing more robust and critical RAG systems is a more promising direction for clinical applications than model specialization alone.

## 1 Introduction

Evidence-Based Medicine (EBM) demands that clinical decisions be grounded in the best available research evidence. The cornerstone of EBM is the systematic review and meta-analysis, which synthesize findings from multiple primary studies to establish clinical guidelines and inform practice (Sackett et al., 1996). However, the volume of medical literature is expanding at an exponential rate, making it practically impossible for clinicians and researchers to manually survey all relevant studies Bornmann et al. (2021). Large Language Models (LLMs) present a promising solution to this information overload, demonstrating an impressive capacity to encode and recall vast amounts of clinical knowledge Singhal et al. (2023); Nori et al. (2023).

The trajectory of medical LLM evaluation has rapidly progressed from foundational benchmarks testing static knowledge on licensing exams Jin et al. (2021); Pal et al. (2022) to more complex assessments of reasoning in simulated clinical environments Kweon et al. (2025); Fan et al. (2025). As model performance on these fact-based tasks approaches saturation Chen et al. (2025b); Tu et al. (2024), the research frontier has shifted toward evaluating more nuanced cognitive skills demanded by real-world clinical practice Arora et al. (2025).

Despite this progress, a critical gap persists. Current benchmarks do not focus on evaluating the core cognitive skill of **multi-source conclusion synthesis**: the ability to analyze findings from multiple, often heterogeneous, primary research articles to construct a coherent, evidence-based conclusion. This skill is fundamental to creating meta-analyses, requiring a model not just to recall facts but to weigh evidence, identify consensus, and abstract novel insights.

To address this gap, we introduce **MedMeta**, the first benchmark designed to evaluate an LLM's ability to perform multi-source conclusion synthesis in a medical context. This benchmark contains 81 curated meta-analyses from PubMed (2018—2025), spanning 24 popular medical specialties.

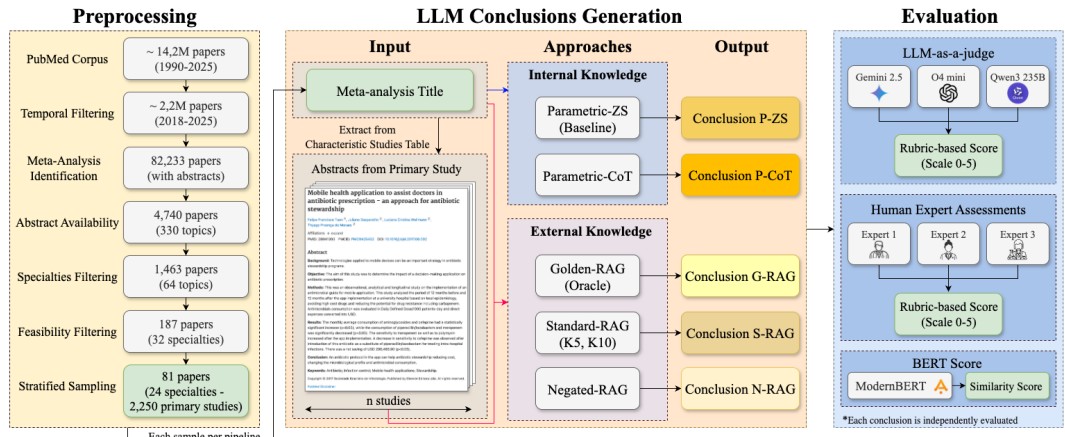

Figure 1: The MedMeta benchmark pipeline. Starting with large-scale filtering of PubMed, meta-analysis studies are identified and screened based on specific inclusion criteria. These are then processed by a set of diverse LLMs under six workflow settings to synthesize conclusions. Final outputs are evaluated using LLM-as-judges, BERTScore, and human expert assessments.

MedMeta challenges models to generate the conclusion of a meta-analysis using only the abstracts of its constituent primary studies. We approach this task using abstracts as a practical proxy for full-text articles, making the task tractable for models with current context window limitations. To specifically test synthesis capabilities under varying conditions, our design includes both parametric and Retrieval Augmented Generation (RAG) workflows. This controlled setup provides models with multiple ground-truth source abstracts and allows us to specifically assess their ability to synthesize information across studies, minimizing the influence of unrelated factors such as document retrieval or LLM context window constraints.

In this work, we make the following contributions:

- We introduce **MedMeta**, a benchmark that evaluates the critical skill of multi-source conclusion synthesis, a cornerstone of evidence-based medicine.

- We validate an **LLM-as-a-judge (LLM-J) protocol**, demonstrating a strong correlation ($r$ up to 0.81) and negligible systematic bias compared to human experts, establishing LLM panels as reliable proxies for evaluating generated conclusions.

- We conduct analyses showing that RAG is more impactful for synthesis quality than domain-specific fine-tuning. Our tests reveal a potentially universal vulnerability in current LLMs, as they often fail to identify and reject factually incorrect evidence.

## 2 RELATED WORK AND BACKGROUND

### 2.1 RETRIEVAL-AUGMENTED GENERATION AND MEDICAL APPLICATIONS

Retrieval-Augmented Generation (RAG) has become the standard paradigm for grounding LLM outputs in external knowledge, mitigating hallucination and enabling access to up-to-date information Lewis et al. (2020). Benchmarks such as RGB Chen et al. (2024b) and RECALL Liu et al. (2023) evaluate retrieval and generation quality in open-domain QA, while frameworks like ARES Saad-Falcon et al. (2024) and CRAG Yan et al. (2024) improve robustness through adaptive retrieval. However, these efforts largely assess fact-finding and conversational QA rather than the abstractive synthesis of a formal scientific conclusion from curated technical documents, which is central to evidence-based medicine (EBM). A further limitation of the RAG paradigm is its susceptibility to noisy or factually incorrect retrievals Zhang & Gao (2024); Fang et al. (2024). Current models often uncritically synthesize such context, failing to cross-check against parametric knowledge or detect internal contradictions Yu et al. (2024); Hong et al. (2024), a vulnerability in medicine implications.

## 2.2 CURRENT MEDICAL BENCHMARKS AND GAPS

Within the medical domain, early evaluations of LLMs have focused on Question Answering. MedQA Jin et al. (2021) and MedMCQA Pal et al. (2022) demonstrated expert-level accuracy on licensing exam questions, while PubMedQA Jin et al. (2019) required binary judgments from single abstracts. These benchmarks confirmed factual recall but did not address the more demanding challenge of synthesizing novel conclusions from multiple heterogeneous studies. Related work has also explored reasoning in clinical settings: EHRNoteQA Kweon et al. (2025) evaluates responses to clinician queries over discharge summaries Johnson et al. (2023), and MedAgentBench Jiang et al. (2025) introduces a virtual EHR environment for task completion. These tasks assess reasoning over a single, coherent clinical document, which differs fundamentally from integrating evidence across multiple, and potentially contradictory, research studies. Other benchmarks emphasize explainability and robustness, such as MedExQA Kim et al. (2024) and related datasets Chen et al. (2025b) that use expert-written rationales, or Med-HALT Pal et al. (2023) and MedXpertQA Zuo et al. (2025) that focus on hallucinations and difficult exam-style questions. While important for reliability, these evaluations do not directly measure generative synthesis.

Despite progressive advances, existing benchmarks share a common limitation. They focus on reasoning over self-contained information (e.g., EHRs), or already-synthesized knowledge (e.g., textbook) rather than on the generative synthesis of new conclusions from primary evidence. The cornerstone of EBM is precisely this cognitive skill, integrating findings from multiple, heterogeneous research articles into a coherent conclusion, yet it remains largely unevaluated. MedMeta addresses this gap by directly benchmarking multi-source conclusion synthesis in the medical domain.

## 3 MEDMETA BENCHMARK

Figure 1 shows the benchmark's design, including three main stages: (1) systematic collection and preprocessing of medical meta-analyses; (2) generation of conclusions using LLM workflows; and (3) an evaluation framework combining automated metrics and human expert assessment.

### 3.1 META-ANALYSIS COLLECTION AND PREPROCESSING

We built a challenging dataset by curating representative meta-analyses from PubMed using a multi-stage filtering pipeline. This ensures each selected study is methodologically sound, not overly well-known, and presents a tractable synthesis task based on abstracts.

**Data Collection.** We initiated the process with a large-scale crawl of 14.2 million papers from the PubMed using E-utilities Sayers (2009). We applied filtering to keep only articles published between 2018 and 2025 to mitigate potential data contamination from the models' pre-training corpora, thereby encouraging evaluation of synthesis rather than retrieval of memorized information.

**Publication Type Filtering.** From this subset, we applied PubMed's built-in publication type filters to identify studies explicitly designated as "meta-analysis" or "systematic review" in their Medical Subject Headings (MeSH) publication type. From the corpus of 2.2 million articles (2018–2025), we identified 82,233 meta-analyses and systematic reviews with full-text availability in PubMed.

**Inclusion Criteria.** To ensure rigor and suitability, a meta-analysis was retained only if it satisfied these conditions: (1) presence of a "Characteristics Studies" table or equivalent structured summary of primary research, (2) all cited primary studies must be retrievable in PubMed with available abstracts and (3) a main conclusion that is sufficiently explicit to be parsed programmatically. This filtering pipeline reduced the candidate pool to 4,740 papers spanning 330 distinct raw topics.

**Specialties Filtering.** Due to the nature of PubMed, authors can freely assign paper categories, making it extremely challenging to maintain consistent topic labels. To address this, we used Gemini Flash 2.5 to process each paper's abstract and title, automatically categorizing them into 64 medical specialties defined by MeSH terms (see Appendix A) and one additional "Other" topic. We then filtered out the "Other" topic, resulting in a diverse and representative set of 1,463 papers covering 64 distinct medical specialties.

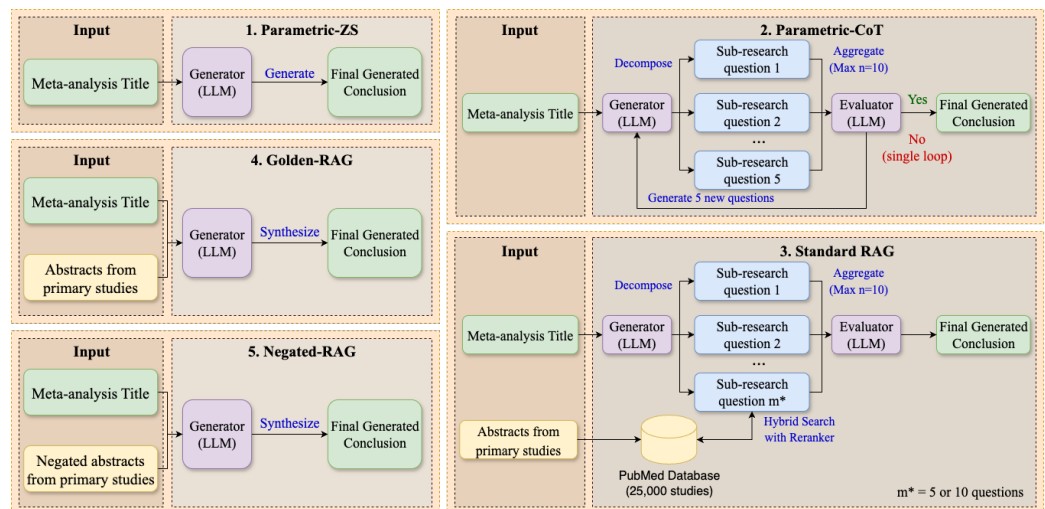

Figure 2: MedMeta workflow architecture. The benchmark includes six distinct synthesis workflows varying in input type (parametric vs. retrieved), reasoning strategy (zero-shot vs. chain-of-thought), and retrieval fidelity (oracle, noisy, or negated), enabling fine-grained evaluation of LLM's performance under different conditions.

**Feasibility Filtering.**  As a final filtering step, we conducted a feasibility check to verify that each meta-analysis's conclusion could be reproduced using only primary study abstracts. This safeguards against missing context, given that full texts are not used, and confirms that LLMs have enough information to synthesize a valid conclusion. Each sample was processed through Gemini Flash 2.5 in three independent runs (temperature 0.5). For each run, we provided the model with the title and conclusion of the meta-analysis, along with the abstracts of all corresponding primary studies. We then asked the model whether the stated conclusion could be reasonably inferred from those abstracts, and averaged its ratings across runs (see Appendix F). Only papers with an average score over 4 were retained. This step was essential to mitigate the inherent information loss from not using full-text articles and to ensure that each task in the benchmark was tractable. This reduced the set to 187 papers. We acknowledge that using an LLM to filter for feasibility could pose a risk of bias. However, the sustained robustness of our results across diverse model families (see Figure 2) suggests that this step did not meaningfully skew outcomes.

**Stratified Sampling.**  To ensure a balanced benchmark, we performed stratified sampling across publication years. This stratification ensures sufficient post-cutoff inputs to mitigate memorization bias in LLMs Carlini et al. (2022). The resulting MedMeta dataset consists of 81 meta-analyses covering 24 medical specialties, with a total of 2,250 primary studies. The complete benchmark is available in our public repository. See benchmark characteristics in Appendix G.

### 3.2 LLM Workflows for Conclusion Generation

To isolate synthesis and retrieval capabilities, we evaluate models across five settings (Figure 2).

**Zero-Shot Baseline (P-ZS).**  This is the simplest workflow, designed to test a model's internal knowledge. The model receives only the title of the meta-analysis and is prompted to directly generate a conclusion (Appendix B). This setting involves no Chain-of-Thought (CoT) prompting Wei et al. (2022), no feedback, and no retrieved context.

**Parametric-CoT (P-CoT).**  This workflow assesses a model's ability to reason with its parametric knowledge through CoT prompting (Appendix C). First, LLM decomposes the meta-analysis title into sub-questions. It answers these questions and aggregates them into a draft conclusion. A feedback loop allows for revision based on new sub-questions if the initial draft is deemed inadequate by an LLM evaluator. This workflow tests structured reasoning without external knowledge.

**Standard-RAG (S-RAG).**    This workflow evaluates RAG performance under realistic noise conditions. Models attempt to synthesize meta-analytic conclusions from document collections containing both relevant and irrelevant content. Following the P-CoT sub-question generation approach, the system retrieves K={5,10} documents per query using hybrid search (BM25 Robertson et al. (1995) and BGE-m3 Chen et al. (2024a)) with BGE-m3-Reranker. We construct a proxy evaluation corpus of 25,000 PubMed abstracts: 2,250 ground-truth abstracts from our 81 meta-analyses alongside 22,750 random noise abstracts (2018–2025). This setup allows examination of retrieval noise effects on synthesis quality.

**Golden-RAG (G-RAG).**    This is an oracle retrieval workflow designed to isolate a model's synthesis capability by eliminating retrieval errors. The model is supplied with the meta-analysis title and the complete set of ground-truth abstracts from all primary studies included in the original meta-analysis (Appendix E; median: 11 abstracts). This oracle configuration provides an upper-bound estimate of synthesis performance under a perfect retrieval condition.

**Negated-RAG (N-RAG).**    To assess model robustness against misinformation, this workflow follows the G-RAG setup but with an adversarial attack: the factual claims within all ground-truth abstracts are systematically negated before being passed to the model (Appendix D). This tests whether models can identify and reject clearly faulty evidence.

**Implementation.**    Workflow orchestration was implemented with LangGraph LangChain (2024), and inference of open-weights models was optimized using vLLM Kwon et al. (2023) (Appendix I). Closed-weights models were accessed via APIs.

## 3.3    EVALUATION FRAMEWORK

**Hypotheses.**    We aim to investigate the following hypotheses on the MedMeta task:

- **H1 (Human vs. LLM-J Alignment):** For the task of evaluating conclusion quality in MedMeta, scores assigned by a panel of LLM-as-a-Judge will show a strong, positive correlation with scores from medical experts.

- **H2 (Information Grounding):** Across all tested models, performance in the RAG workflow will be significantly higher than in the Parametric workflow.

- **H3 (Domain Adaptation):** For our selected model pair (Gemma and MedGemma), the domain-specialized model will outperform its general-purpose counterpart, with this effect being most pronounced in knowledge-intensive, non-RAG settings.

**Automated Evaluation Metrics.**    Evaluating abstractive summaries at scale requires robust automated metrics. Recent studies show that large LLMs can approximate human judgment with both scalability and consistency Zheng et al. (2023); Chiang & Lee (2023). Following this paradigm, we employ an LLM-J panel composed of three frontier models (Gemini 2.5 Pro, O4 mini, and Qwen3 235B). Each model scores generated conclusions against reference conclusions using a detailed rubric (Appendix E.2), with temperature fixed at 0.0 and reasoning mode enabled. Final scores are averaged across judges, reducing individual bias and improving robustness. As a complementary metric, we also compute semantic similarity using BERTScore Zhang et al. (2020).

**Human Expert Validation.**    To validate our automated metrics (H1), we recruited nine annotators with medical backgrounds (see Table 1). We randomly subsampled 20 meta-analyses, and used a Latin Square design to minimize bias Fisher (1935). Each generated conclusion was independently scored by three annotators on the same rubric used by the LLM panel. All annotators had to complete a training session before beginning the evaluation tasks (Appendix H).

**Statistical Analysis.**    We assess alignment between human and automated judges (H1) using Pearson correlation Pearson (1895) for linear relationships and Bland-Altman analysis Bland & Altman (1986) to examine absolute agreement and systematic bias. To test hypotheses (H2) and (H3), we apply paired t-tests Student (1908) to evaluate the statistical significance of performance differences.

| Background | Count | Years of Experience |
|---|---|---|
| Pharmacists | 3 | $1.6 \pm 0.36$ |
| Biologists | 2 | $1.8 \pm 0.71$ |
| Biohealth (Master's) | 4 | $2 \pm 0$ |

Table 1: Human Expert Annotator Profiles

**Model Selection.** We evaluate a diverse set of leading open and closed-weights models across model size. This includes Gemini Flash 2.5, O4 Mini, several 8B models from the Qwen family with and without native CoT reasoning OpenAI et al. (2024), and the 27B Gemma/MedGemma pair, allowing for a broad overview of current model capabilities on the synthesis task.

For our targeted hypothesis testing, we focus on Gemma and its medical derivative, MedGemma. This choice is twofold. First, their shared architecture provides a controlled setting to isolate the effects of domain-specific fine-tuning. Second, given the resource-intensive nature of recruiting and training annotators with medical expertise, concentrating our human validation study on this single, controlled pair allowed for a rigorous yet feasible validation of our evaluation framework.

## 4 RESULTS

Our comprehensive evaluation, summarized in Table 2, yields three primary conclusions. First, information grounding is essential. RAG-based workflows consistently outperform parametric approaches across all models, establishing access to evidence as the most critical factor for high-quality synthesis. Second, the benefits of domain adaptation are modest and depend on context. The advantage of the specialized MedGemma model becomes negligible once external evidence is provided through RAG. Third, we uncover a universal vulnerability across current architectures. All models, regardless of size or specialization, fail our adversarial test by uncritically incorporating misinformation into outputs that are coherent but factually false.

**The Value of Structured Reasoning.** A consistent observation across all models is the performance gain achieved through simple prompting techniques. The P-CoT workflow, which introduces a CoT structure with a feedback loop, consistently outperforms the P-ZS baseline ($\sim$30-33%). This suggests that, even without external evidence, prompting the model to decompose the problem into smaller steps gives it more room to reason Chen et al. (2025a). This process helps the model better utilize its internal knowledge, expanding its effective search space and improving its ability to generate coherent and relevant conclusions.

**The Impact of Retrieval.** Introducing external evidence via RAG yields a significant improvement in synthesis quality. Across all models, RAG workflows consistently score higher than P-CoT methods. This performance uplift is substantial and varied, ranging from a $\sim$9% increase for Gemini Flash 2.5 to over 40% for the Gemma models, underscoring the critical benefit of grounding over relying on a model's internal knowledge alone.

**Robustness of Frontier Models to Noisy Retrieval.** Our results indicate that the optimal amount of retrieved context is not universal but depends on model capability. As shown in Table 2, there is no consistent winner between the Standard-RAG (K=5) and (K=10) workflows. More capable models like Gemini Flash 2.5 and O4 Mini appear to benefit from a larger context (K=10), suggesting they can effectively sift through more documents to find relevant evidence. Conversely, other models show comparable or slightly better performance with a more focused context (K=5). This suggests a practical trade-off for these models, where the risk of introducing distracting information with a larger context may outweigh the benefit of potentially higher recall.

**Trade-Offs Between Context Size and Model Capability.** Another finding is that the performance penalty for imperfect retrieval is minimal for larger models. For Gemini Flash 2.5 and O4 Mini, the performance of Standard-RAG is statistically indistinguishable from the oracle G-RAG setting. This result suggests that these advanced models, when paired with a strong reranker, are

| Model | P-ZS | P-CoT | G-RAG | K5-RAG | K10-RAG | N-RAG |
|---|---|---|---|---|---|---|
| Gemini Flash 2.5 | 2.10 | 2.90 ± 0.21 | 3.16 ± 0.18 | 3.03 ± 0.16 | **3.17 ± 0.18** | 1.01 ± 0.19 |
| O4 Mini | 2.00 | 2.70 ± 0.22 | 2.79 ± 0.20 | 2.90 ± 0.18 | **2.94 ± 0.21** | 1.19 ± 0.24 |
| MedGemma 27B | 1.80 | 2.17 ± 0.21 | **2.72 ± 0.17** | 2.68 ± 0.19 | 2.46 ± 0.20 | 1.00 ± 0.18 |
| Gemma 27B | 1.60 | 1.77 ± 0.22 | **2.58 ± 0.20** | 2.37 ± 0.22 | 2.31 ± 0.22 | 0.98 ± 0.19 |
| Qwen3 8B | 1.70 | 2.27 ± 0.24 | **2.72 ± 0.16** | 2.53 ± 0.24 | 2.63 ± 0.22 | 1.03 ± 0.20 |
| Qwen3 8B (reasoning) | 1.50 | 2.00 ± 0.22 | **2.56 ± 0.19** | 2.46 ± 0.22 | 2.30 ± 0.25 | 1.00 ± 0.19 |
| Qwen3 8B-DeepSeek | 1.30 | 1.94 ± 0.24 | **2.55 ± 0.16** | 2.10 ± 0.24 | 2.13 ± 0.25 | 1.17 ± 0.23 |

Table 2: Mean LLM-Judge scores (±95% CI) across models and retrieval settings. Scores are on a 0–5 scale with 5 is the highest. Bold values indicate the best-performing workflow for each model.

capable of identifying the most salient evidence from a noisy retrieval set, effectively matching the performance of a system with perfect recall. For the other models, however, a performance gap remains between Standard-RAG and G-RAG, indicating their synthesis quality is more fundamentally constrained by the precision of the retrieval step.

**Vulnerability to Misinformation.** The N-RAG performance reveals a critical common failure to all tested models. Despite being provided with factually inverted and contradictory information, every model proceeded to synthesize these incorrect claims into a coherent but false conclusion. The resulting scores are significantly lower than even the "P-ZS" baseline. Particularly, this is striking for more capable models like Gemini Flash 2.5 and O4 Mini, which might be expected to leverage their extensive parametric knowledge to detect such contradictions but fail to do so. This finding empirically confirms the vulnerability of RAG systems to faulty evidence and demonstrates that current models act as obedient synthesizers rather than critical reasoners, lacking the capability to identify and reject misinformation based on internal knowledge or logical inconsistency.

**Task-Dependent Efficacy of Reasoning Modes.** Analysis of the Qwen models, which offer an explicit "reasoning" mode, indicates that the utility of such features may be task-dependent. We did not find a consistent performance gain from this mode compared to the standard instruction-tuned variant; for the P-CoT workflow, scores were slightly lower (Table 2). This result contrasts with the well-documented benefits of general CoT prompting for complex reasoning problems Wei et al. (2022). A plausible explanation for this discrepancy is the nature of our constrained synthesis task. For tasks that primarily require abstracting and rephrasing provided information, a direct instruction-following approach may be more robust. The addition of deliberative reasoning steps could introduce processing artifacts or cause deviations from the core synthesis objective.

## 5 MANUAL ANALYSIS

### 5.1 VALIDATION OF THE LLM-J PROTOCOL (H1)

A prerequisite for the large-scale analysis in this study is a reliable automated evaluation metric. To this end, we validated our LLM-J protocol against human medical experts. We computed correlation and reliability metrics between mean LLM judge scores (n=3) and human annotator scores (n=3) on 20 samples per condition. Strong Pearson correlations emerged across all models and workflows ($r$ = 0.65-0.81, $p < 0.01$; Table 3), demonstrating a positive relationship human experts and LLMs.

Although a high correlation (Pearson's $r$) indicates association, it does not imply interchangeability Novikova et al. (2017); Sellam et al. (2020). We therefore applied Bland–Altman analysis, a standard method in clinical research for comparing two measurement techniques. For each generated conclusion $i$, scored by humans ($S_{H,i}$) and LLMs ($S_{L,i}$), we computed the difference $d_i = S_{H,i} - S_{L,i}$. The analysis focuses on two key values: the mean bias ($\bar{d}$), representing systematic difference, and the 95% limits of agreement (LoA).

$$\text{Mean Bias} = \bar{d} \qquad \text{LoA} = \bar{d} \pm 1.96 \times \text{SD}(d) \tag{1}$$

where $\text{SD}(d)$ is the standard deviation of the differences.

| Model & Workflow | Pearson's $r$ | Mean Bias | 95% LoA |
|---|---|---|---|
| MedGemma 27B (Golden-RAG) | 0.74 | +0.14 | [-1.18, 1.46] |
| Gemma 27B (Golden-RAG) | 0.65 | +0.31 | [-1.42, 2.04] |
| MedGemma 27B (Parametric-CoT) | 0.81 | +0.27 | [-1.08, 1.62] |
| Gemma 27B (Parametric-CoT) | 0.70 | +0.10 | [-1.58, 1.78] |

Table 3: Human–LLM-J alignment (H1). Correlation and bias with human expert scores.

We observe 2 key results that strongly support (H1). First, the mean bias is consistently close to zero across all settings and is not statistically significant (Paired t-tests, all $p > 0.10$), indicating no systematic tendency for the LLM-J to score higher or lower than human experts. Second, the 95% LoA provide a clinically interpretable range of expected error. For instance, in MedGemma G-RAG, the LoA of [-1.18, 1.46] means that for any given conclusion, the LLM score is expected to be within approximately 1.5 points of the human score 95% of the time on 0–5 scale. A qualitative review of the Bland-Altman distribution showed that the differences between human and LLM-J scores were scattered evenly around the mean bias across the range of scores, indicating that the level of agreement does not systematically vary with the quality of the conclusion being evaluated. The strong correlation, negligible systematic bias, and well-defined LoA provide robust evidence that our LLM-J protocol can serve as a valid and reliable proxy for human experts in evaluating conclusions within MedMeta.

## 5.2 THE ROLE OF INFORMATION GROUNDING (H2)

| Hypothesis | Comparison | Judge | N | Mean Diff. | t-stat | p-value | Cohen's d | Sig. |
|---|---|---|---|---|---|---|---|---|
| *H2: Information Grounding (G-RAG vs. P-CoT)* | | | | | | | | |
| | MedGemma 27B | Human | 20 | 0.742 | 2.314 | 0.032 | 0.517 | Yes |
| | MedGemma 27B | LLM | 81 | 0.543 | 4.347 | 0.001 | 0.483 | Yes |
| | Gemma 27B | Human | 20 | 1.025 | 3.289 | 0.004 | 0.735 | Yes |
| | Gemma 27B | LLM | 81 | 0.807 | 5.745 | 0.001 | 0.638 | Yes |
| *H3: Domain Adaptation (MedGemma vs. Gemma)* | | | | | | | | |
| | Golden-RAG | Human | 20 | 0.150 | 1.084 | 0.292 | 0.242 | No |
| | Golden-RAG | LLM | 81 | 0.140 | 1.952 | 0.054 | 0.217 | No |
| | Parametric-CoT | Human | 20 | 0.433 | 1.317 | 0.203 | 0.295 | No |
| | Parametric-CoT | LLM | 81 | 0.403 | 2.935 | 0.004 | 0.326 | Yes |

Table 4: Paired t-test results for H2 (Information Grounding: G-RAG vs. P-CoT) and H3 (Domain Adaptation: MedGemma-27B vs. Gemma-27B). "Yes" indicates significance at $\alpha = 0.05$.

Consistent with prior work on RAG Lewis et al. (2020); Gao et al. (2023), we established the baseline effect of providing evidence, hypothesizing that grounding models in abstracts would yield higher-quality conclusions. We therefore hypothesized that this principle would hold true in our challenging MedMeta setting, which requires long-context synthesis: that grounding models in large amounts of ground-truth abstracts would still lead to significantly higher-quality conclusions than relying on parametric knowledge alone.

Our results provide clear evidence for H2 (Table 4), demonstrating the critical role of information grounding in medical conclusion synthesis. In all tested conditions, conclusions generated via the G-RAG workflow were rated as significantly superior to those from the P-CoT approach (Table 4). This finding was robust across both human and LLM judges, with all comparisons yielding statistical significance ($p < 0.04$) and medium-to-large effect sizes (Cohen's $d = 0.48$ to $0.74$). The magnitude of this improvement was notable; for the general-purpose Gemma model, human judges rated RAG-based conclusions over a full point higher on a five-point scale (Mean Diff = 1.025). This consistent and substantial performance gain confirms that providing access to ground-truth evidence is a primary determinant of synthesis quality, establishing a clear baseline for the subsequent analysis of domain adaptation.

## 5.3 THE BENEFITS OF DOMAIN ADAPTATION (H3)

Our analysis for H3 reveals that the benefits of domain-specific fine-tuning are likely modest and context-dependent. The advantage of the specialized MedGemma model is largely neutralized when RAG provides external material. In the G-RAG setting, we observed no statistically significant performance difference between MedGemma and Gemma, as evaluated by either experts or LLM judges ($p > 0.05$ for both). Small effect sizes (Cohen's $d \leq 0.25$) indicate that general-purpose models can perform on par with specialized fine-tuned ones when sufficiently grounded.

In contrast, an advantage for MedGemma emerges in the P-CoT setting, where it relies solely on internal knowledge. Here, LLM judges rated MedGemma's conclusions as significantly higher quality than Gemma's (Mean Diff = 0.403, $p = 0.004$). This pattern suggests that domain adaptation primarily enhances the recall and structuring of parametric knowledge. For complex synthesis tasks like MedMeta, these findings indicate that investing in high-quality retrieval systems may offer a greater return than specializing models through fine-tuning.

## 6 INSUFFICIENCY OF BERTSCORE SIMILARITY METRICS

We further investigated whether a standard automated metric (BERTScore) could serve as a reliable proxy for evaluating conclusion quality. Our hypothesis was that semantic similarity alone would be insufficient to capture the factual and logical nuances of synthesis. The results, shown in Table 5, confirm this hypothesis. BERTScore fails to differentiate workflows, giving nearly identical high F1 scores across them. Most critically, it rates false conclusions from the N-RAG semantic equivalent to G-RAG. High token-level semantic overlap provides a poor proxy for factual accuracy, since a generated conclusion can appear highly similar to a reference text while remaining factually incorrect or critically flawed. These findings support our use of rubric-based LLM-J protocol.

| Model | Parametric-CoT | Golden-RAG | Negated-RAG |
|---|---|---|---|
| Gemini Flash 2.5 | $0.850 \pm 0.010$ | $0.855 \pm 0.011$ | $0.845 \pm 0.012$ |
| O4 Mini | $0.830 \pm 0.011$ | $0.835 \pm 0.010$ | $0.840 \pm 0.011$ |
| Gemma | $0.845 \pm 0.012$ | $0.840 \pm 0.012$ | $0.840 \pm 0.013$ |
| MedGemma | $0.840 \pm 0.010$ | $0.840 \pm 0.011$ | $0.843 \pm 0.010$ |
| Qwen3 8B | $0.835 \pm 0.011$ | $0.840 \pm 0.010$ | $0.845 \pm 0.012$ |
| Qwen3 8B (reasoning) | $0.840 \pm 0.012$ | $0.840 \pm 0.011$ | $0.845 \pm 0.011$ |
| Qwen3 8B DeepSeek | $0.835 \pm 0.013$ | $0.840 \pm 0.012$ | $0.842 \pm 0.013$ |

Table 5: Mean BERTScore F1 ($\pm$ standard error) across models and evaluation approaches.

## 7 CONCLUSION

In this work, we introduced MedMeta benchmark to evaluate the critical yet under-explored capability of multi-source conclusion synthesis in medicine. We successfully validated an LLM-J protocol, demonstrating strong alignment with experts and establishing it as a reliable, scalable proxy for evaluating medical conclusions. Our findings reveal a clear hierarchy of importance. Information grounding (RAG) provides a larger performance uplift than domain-specific fine-tuning. Our stress tests demonstrate that surface-level similarity metrics (BERTScore) are inadequate and that current LLMs universally fail to reject factually incorrect evidence.

While the results are promising, several limitations point to future directions. First, using abstracts as a proxy for full-texts may miss study nuances. Second, human validation was limited to 9 expert annotators and focused on a subset of models and settings. Third, expanding beyond 81 meta-analyses and 24 specialties would further enhance its comprehensiveness. Finally, as LLMs evolve rapidly, future work should extend this analysis to novel architectures (MoE) and paradigms (Agents).

The MedMeta benchmark lays a foundation for future inquiry into automated scientific reasoning. Our next steps include expanding to full-text synthesis to capture study nuances, performing multilingual evaluations to assess cross-linguistic synthesis capabilities, and building models with stronger critical reasoning to resist incorrect factual.

## ETHICS STATEMENT

This work does not involve human subjects, personally identifiable information, or sensitive data. The experiments are conducted exclusively on publicly available benchmark datasets under their respective licenses. The proposed methods do not present foreseeable risks of harm, misuse, or unfair discrimination. We adhere to the ICLR Code of Ethics and confirm compliance with standard practices regarding data handling, reproducibility, and research integrity.

## REPRODUCIBILITY STATEMENT

We have made every effort to ensure the reproducibility of our results. The main paper describes the architecture of the MedMeta workflow and the evaluation protocols (Section 3). Detailed inference parameters, prompts, and additional experimental results are provided in the Appendix. To facilitate reproducibility, we will release the source code and scripts in an anonymous repository during the review process. The repository will include: (i) a `Data` folder containing the preprocessed MedMeta dataset, (ii) a `Scripts` folder with step-by-step scripts for reproducing experiments, (iii) a `Src` directory with LangGraph implementations, and (iv) a `Web` folder containing the source code of the annotation platform. A comprehensive README file with setup instructions, dependencies, and usage examples will also be provided.

## LLM USAGE

In this paper, LLMs were used solely as an assistive tool to improve the clarity and readability of the manuscript text. No part of the research ideation, methodology, experimental design, or analysis relied on LLMs. The authors take full responsibility for the content of this work.

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

# APPENDIX

## A MESH TOPIC DISTRIBUTION IN THE MEDMETA

To ensure the breadth and clinical relevance of our MedMeta benchmark, we curated tasks spanning the major branches of the MeSH taxonomy. This diverse sampling strategy, detailed below, validates our benchmark's comprehensiveness and tests the generalization capability of the evaluated models across distinct medical domains. This approach miti-

gates the risk of our benchmark being biased towards a narrow set of medical fields.

---

**Anatomy [A]:** Body Regions [A01], Musculoskeletal [A02], Digestive [A03], Respiratory [A04], Urogenital [A05], Endocrine [A06], Cardiovascular [A07], Nervous [A08], Sense Organs [A09], Tissues [A10], Cells [A11], Fluids and Secretions [A12], Animal Structures [A13], Stomatognathic System [A14], Hemic/Immune [A15], Embryonic Structures [A16], Integumentary System [A17], Plant Structures [A18], Fungal Structures [A19], Bacterial Structures [A20], Viral Structures [A21].

**Diseases [C]:** Infections [C01], Neoplasms [C04], Musculoskeletal Dis. [C05], Digestive Dis. [C06], Respiratory Dis. [C08], Otorhinolaryngologic [C09], Nervous Dis. [C10], Eye Diseases [C11], Urogenital Diseases [C12], Cardiovascular Dis. [C14], Hemic and Lymphatic [C15], Congenital [C16], Skin Diseases [C17], Metabolic Dis. [C18], Endocrine Dis. [C19], Immune Dis. [C20], Disorders [C21], Animal Diseases [C22], Pathological Conditions [C23], Occupational Diseases [C24], Chemically-Induced [C25], Wounds and Injuries [C26].

**Chemicals & Drugs [D]:** Pharmaceutical Prep. [D26], Inorganic Chemicals [D01], Organic Chemicals [D02], Heterocyclic Compounds [D03], Polycyclic Compounds [D04], Macromolecular [D05], Hormones [D06], Enzymes and Coenzymes [D08], Carbohydrates [D09], Lipids [D10], Amino Acids [D12], Nucleic Acids [D13], Complex Mixtures [D20], Biological Factors [D23], Biomedical Materials [D25], Pharmaceutical [D26], Chemical Actions [D27].

**Techniques [E]:** Diagnosis [E01], Therapeutics [E02], Anesthesia and Analgesia [E03], Surgical Procedures [E04], Investigative Techniques [E05], Dentistry [E06], Equipment and Supplies [E07].

**Psychology [F]:** Behavior [F01], Psychological Phenomena [F02], Mental Disorders [F03], Behavioral Disciplines [F04].

---

# B    PROMPTS FOR LLM ZERO-SHOT

This is the baseline prompt for LLM to use its own knowledge to create meta-analysis conclusion.

---

You are an expert Clinical Research Scientist specializing in evidence-based medicine and the interpretation of meta-analyses. Your primary skill is to synthesize complex medical information into clear, concise conclusions.
Your task is to generate the most likely primary concluding statement for a medical meta-analysis, based solely on its title (research question).
You will be provided with only the following information: Meta-Analysis Title: `[Meta-Analysis Title]`
Core Instructions: 1. You must provide the best possible conclusion based on the title and your existing knowledge. 2. The conclusion should be a single, concise, and coherent conclusion paragraph.
Provide your response as a single block of text containing only the generated conclusion. Do not include any preceding or succeeding conversational text, introductions, or apologies.

---

# C    PROMPTS FOR LLM WORKFLOWS

This section details the sequence of prompts used in our proposed workflows. Each prompt is engineered to elicit a specific cognitive task from the LLM, breaking down the complex process of meta-analysis conclusion generation into a structured, multi-step reasoning process. Full prompt details are available in our public repository.

## C.1    PROMPT FOR DECOMPOSING THE RESEARCH TOPIC

This initial prompt bootstraps the process by instructing the LLM to structure the research problem into a coherent plan, including key research questions.

---

You are a research assistant skilled in formulating structured research plans for systematic reviews or meta-analyses. Given a research topic, create a concise plan including background context, 5 key research questions, and a brief summary of the search strategy/concepts.

---

## C.2 PROMPT FOR INITIAL KNOWLEDGE GENERATION (PARAMETRIC-CoT)

This prompt queries the LLM's internal knowledge base to generate an initial, comprehensive answer to the primary research question.

> You are an **expert researcher with broad medical knowledge**. For the given research question, provide a comprehensive answer based on your internal knowledge. If applicable, identify 2-3 critical sub-questions that arise from this research question and provide detailed answers to those as well within your response. Structure your entire response as a single coherent text.

## C.3 PROMPT FOR FEEDBACK INTEGRATION

This prompt guides the LLM to evaluate its own initial output against the research plan, identify gaps, and generate new, targeted questions to address shortcomings.

> You are an expert research evaluator tasked with assessing whether a generated conclusion adequately addresses and matches the given research topic. Your evaluation should consider:
> 1. **Topic Relevance**:
> 2. **Comprehensiveness**:
> 3. **Specificity**:
> 4. **Coherence**:
> 5. **Completeness**:
> Provide your assessment as:
> 1. A detailed evaluation explaining what works well and what might be missing or inadequate
> 2. A score from 0-5 where:
> - 0 = Completely inadequate
> - 1 = Very inadequate
> - 2 = Inadequate
> - 3 = Moderately adequate
> - 4 = Good
> - 5 = Excellent
> Focus on whether the conclusion is sufficient for someone researching this specific topic.
> Research Topic: [Topic]
> Current Research Plan: [Context]
> Generated Conclusion: [Conclusion]
> Please evaluate whether this conclusion adequately matches and addresses the research topic.
> Provide both a detailed evaluation and numerical score 0-5.

### FEEDBACK-DRIVEN QUESTION REFINEMENT PROMPT

The agent is prompted to formulate a *new set of research questions*. This crucial step operationalizes the feedback, guiding the agent to explicitly target the identified knowledge gaps in the next iteration of answer generation.

> You are a research assistant expert at formulating targeted research questions. Given a research topic, original questions that were already asked, and feedback about what was missing from the initial conclusion, generate 5 NEW and DIFFERENT sub-questions that will help address the gaps and improve understanding of the research topic.
> Your new questions should:
> 1. Be completely different from the original questions
> 2. Address specific gaps mentioned in the evaluation feedback
> 3. Explore different angles, perspectives, or aspects of the topic
> 4. Be specific and actionable for research purposes
> 5. Help fill in missing information to better address the research topic
> Research Topic: [Topic]
> Original Questions Already Asked: [Previous Research Questions]
> Evaluation Feedback (what was missing/inadequate): [Evaluation Feedback]
> Generate 5 NEW sub-questions that are different from the original ones and will help address the gaps identified in the evaluation feedback:

## C.4 Prompt for Synthesizing the Final Conclusion

Used in both workflows, this final prompt instructs the LLM to distill all available context (either from its internal reasoning or retrieved abstracts) into a single, focused concluding statement.

> You are a research analyst tasked with drafting the **primary concluding statement** for a meta-analysis or systematic review. Your goal is to distill the provided context into the **single most important and specific takeaway message**, as if you were presenting the main result of the study.
> Based **strictly** on the provided context:
> 1. Identify the **central, affirmative findings** or **key definitive statements** made. What is the most crucial outcome, comparison, or result reported?
> 2. Capture any **critical quantifications, effect sizes, or specific comparisons** that are central to this main finding.
> 3. Include any **essential caveats, limitations, or conditions** that are directly tied to and qualify this primary finding.
> 4. The conclusion should be **highly focused and concise**, reflecting the punchline of the research. Avoid general summaries of the entire field or background information from the context.
> 5. Do not introduce external knowledge or comment on the completeness of the provided context.
> Research Topic: [Topic]
> Primary Abstracts: [Context]
> Synthesize the primary concluding statement based **only** on the provided context, focusing on the most direct and impactful findings:

## D Prompt for Negating Facts

This is the prompt using LLM to negate facts in the original meta-analysis conclusion

> You are a medical research assistant. Your task is to create a negated/opposite version of the given meta-analysis text while maintaining scientific credibility and plausibility.
> Given the following meta-analysis title and abstract, create a similar text but with conclusions that are opposite or contradictory to the original. Make sure to: 1. Keep the same title format and structure 2. Maintain the same study design and methodology description 3. Change only the findings/conclusions to be opposite or contradictory 4. Ensure the negated conclusion is medically plausible and realistic 5. Use appropriate medical terminology and maintain scientific rigor
> Original text: [Original Conclusion]
> Create a negated version with opposite conclusions:

## E Evaluation Framework and Rubric

To ensure that our evaluation was rigorous, consistent, and reproducible, we developed an evaluation framework. This framework was applied uniformly to both our LLM-J and human expert evaluations, strengthening the validity of our comparative results.

### E.1 LLM-J Prompt

To standardize our automated evaluation, we designed a detailed prompt that constrains the LLM-J. This prompt establishes a clear expert persona, defines the evaluation task precisely, and provides structured instructions to ensure consistent and criteria-driven assessments. The key components of the prompt are excerpted below. The full prompt is available at our repository,

**Persona & Objective:** You are an expert **Clinical Research Scientist and Critical Appraiser** specializing in meta-analysis methodology and scientific communication.
**Input Data:** You will receive:
1. `[Generated Conclusion]`;
2. `[Original Conclusion]`.
**Core Task:** Evaluate the `[Generated Conclusion]` based on its semantic alignment and completeness compared to the `[Original Conclusion]`.
**Scoring Rubric (0-5 Scale):**
[...]
**Instructions for Evaluation:**
[...]
**Evaluation Criteria:** Focus on the semantic meaning and core components typically found in meta-analysis conclusions. [...]
**Output Format:**
1. Justification: [Your detailed explanation]
2. Score: [Your score from 0-5]

### E.2 SEMANTIC EQUIVALENCE EVALUATION RUBRIC

To ensure both human and LLM evaluators applied consistent standards, we developed the following detailed rubric. This rubric operationalizes the concept of "conclusion quality" into 5 measurable dimensions, focusing on semantic equivalence and the preservation of critical components from the original text. It provided a calibrated scale for all annotations, enhancing the reliability of our results.

**Evaluation Criteria:** Focus on semantic meaning and core components across:
(1) *Main Finding(s)/Overall Result* - primary outcomes;
(2) *Key Specifics & Comparisons* - quantitative results;
(3) *Nuance & Limitations* - caveats, research needs;
(4) *Implications & Future Directions* - clinical significance;
(5) *Safety/Tolerability* - adverse effects if applicable;
(6) *Overall Semantic Equivalence* - core message preservation.
**Scoring Rubric (0-5):**
**5** = Excellent Equivalent (all criteria met);
**4** = High Equivalent (main findings + most specifics);
**3** = Moderate Equivalent (main findings but missing details);
**2** = Low Related (some elements, misrepresents core);
**1** = Very Low Related (substantially different);
**0** = Contradictory.

# F PROMPT FOR RAG FEASIBILITY CHECK

A key methodological concern for any RAG system is whether the retrieved context contains sufficient information to complete the task. To address this, we conducted a "feasibility check" to quantify the information ceiling for our RAG models. The prompt below was used to have an LLM-evaluator determine if the ground-truth conclusion could be reasonably reconstructed from

the provided abstracts alone, helping us interpret the performance of our RAG-based systems.

> Your task is to assess if someone could reasonably arrive at the same conclusion as the original authors by reading only the provided abstracts.
> Provide your assessment as:
> 1. A detailed evaluation including:
> - What key information from the original conclusion is present in the abstracts
> - What important information from the original conclusion might be missing
> - Whether the abstracts provide sufficient evidence to support the original conclusion
> - Any gaps or limitations that would prevent recreating the original conclusion
> 2. A score from 0-5 where
> - 0 = Completely insufficient
> - 1 = Very insufficient
> - 2 = Insufficient
> - 3 = Moderately sufficient
> - 4 = Good sufficiency
> - 5 = Excellent sufficiency
> Focus specifically on whether the abstracts support the original conclusion's claims, findings, and recommendations.
> Research Topic: [Topic]
> Original Conclusion (to be recreated): [Original Conclusion]
> Primary Abstracts: [List of Abstracts]
> Provide both a detailed evaluation and numerical score (0-5).

## G  BENCHMARK CHARACTERISTIC

### G.1  TOPIC DIVERSITY OF BENCHMARK DATA

To demonstrate the breadth of our benchmark, Figure 3 presents the distribution of the most frequent research specialties within MedMeta. This diversity ensures that our evaluation is comprehensive and not limited to a narrow medical domain, thereby testing the generalizability of the models against varied terminologies and concepts.

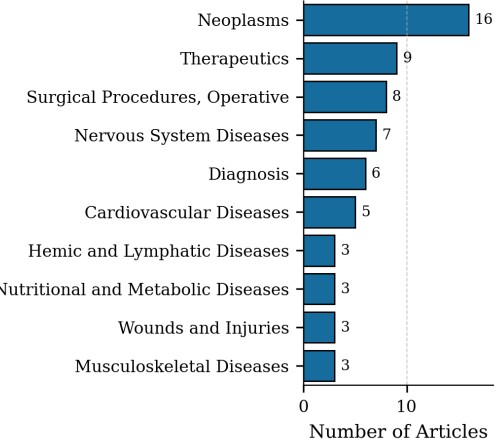

Figure 3: Distribution of the top 10 research specialties in the MedMeta benchmark.

### G.2  COMPLEXITY OF BENCHMARK SOURCE ARTICLES

To characterize the complexity of the source documents, Figure 4 illustrates the distribution of reference counts per meta-analysis. The right-skewed distribution, with a notable median, indicates that our benchmark includes articles with a wide range of scopes—from concise reviews to highly comprehensive analyses.

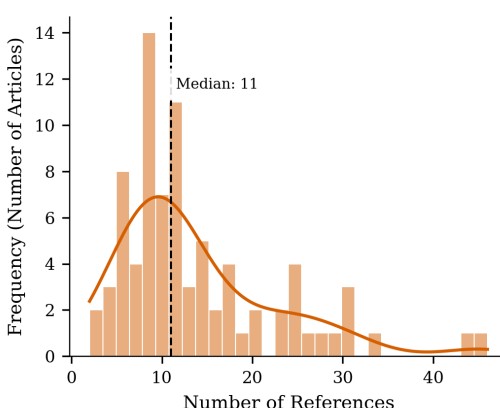

Figure 4: Distribution of reference counts in the source articles of the MedMeta benchmark.

### G.3 YEAR DISTRIBUTION OF BENCHMARK DATA

To confirm the temporal robustness of our benchmark, Figure 5 shows the publication year distribution of the source meta-analyses. The distribution spans over 8 years, ensure evaluating a model's ability to synthesize information from studies with varying reporting styles and terminologies over time.

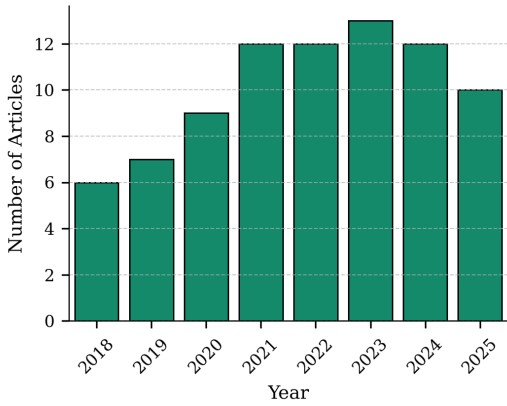

Figure 5: Publication year distribution of source articles in the MedMeta benchmark.

## H    HUMAN ANNOTATION PROTOCOL AND PLATFORM

To create a high-quality ground truth, we designed a multi-stage annotation protocol supported by a custom platform, aimed at maximizing consistency, minimizing bias, and capturing nuanced human judgments.

### H.1    ONBOARDING AND COMMITMENT

Annotators began with an onboarding screen (Figure 6), where they provided email credentials and formally committed to completing all assigned tasks, establishing accountability and engagement.

### H.2    DETAILED SCORING RUBRIC

Each annotator used a detailed 0-5 rubric (Figure 7) with clear qualitative anchors from "No Similarity" to "Excellent Similarity" to assess factual accuracy, main findings, and nuance.

### H.3 ANNOTATOR TRAINING AND CALIBRATION

Before evaluation, annotators completed a calibration phase with gold-standard examples (Figure 8). Highlighted justifications and correct scores helped align annotator judgments to the rubric.

### H.4 LIVE ANNOTATION INTERFACE

During the main task (Figure 9), annotators reviewed two anonymized model-generated conclusions against a reference and scored each using the rubric. A structured interface with progress tracking supported consistent and unbiased annotation.

## I INFERENCE AND COMPUTATION SETUP

Inference for open-weights models was conducted on a local cluster equipped with NVIDIA 4xA6000 48GB GPUs. We utilized the vLLM library (v0.8.3) for efficient deployment. For the 27B parameter models (MedGemma and Gemma), we employed a configuration of 2-way tensor parallelism and 2-way data parallelism, with a maximum context length of 64,000 tokens.

For the Qwen 8B model family, we followed the official guidelines for vLLM deployment. The standard and reasoning variants were configured with 2-way tensor and 2-way data parallelism and a 64,000 token context length, with the "enable-thinking" flag set to "false" and "true", respectively. We deploy DeepSeek Qwen 8B variant with the same parallel and context length configuration.

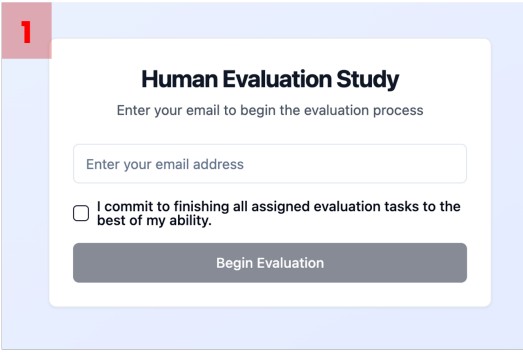

Figure 6: The initial onboarding screen where annotators commit to the evaluation process.

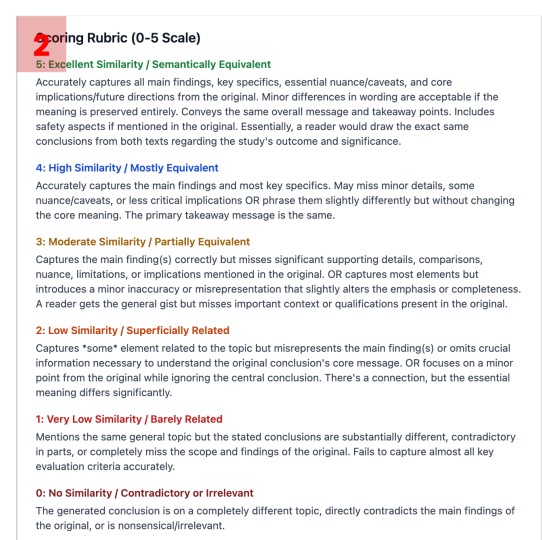

Figure 7: The detailed 0-5 scoring rubric provided to all annotators.

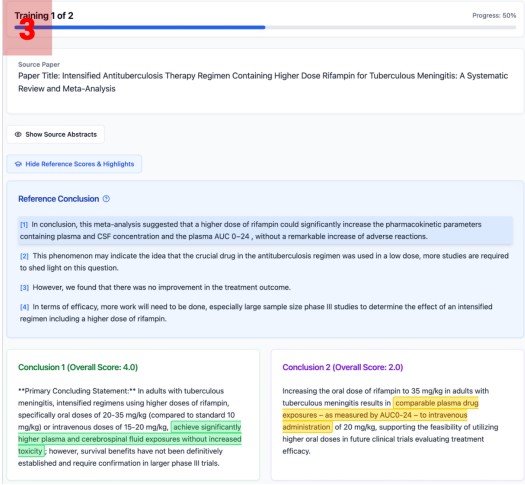

Figure 8: The training interface showing a pre-scored example with highlighted justifications to calibrate annotators.

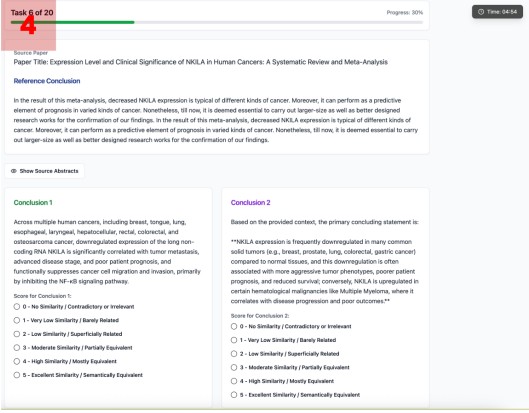

Figure 9: The live annotation interface where annotators evaluate two anonymized model conclusions against the reference conclusion.

