# OpenReview forum: "MedMeta: A Benchmark for LLMs in Synthesizing Meta-Analysis Conclusion from Medical Studies"
_ICLR.cc/2026/Conference — Submitted to ICLR 2026_

### Official Review · Reviewer_s3go · 2025-10-19

**Soundness:** 2
**Presentation:** 3
**Contribution:** 3
**Rating:** 4
**Confidence:** 4

**Summary:**

The paper introduced a benchmark to evaluate the LLM's ability to draw conclusions based on medical papers. The benchmark is built based on existing benchmarks from PubMed. Multiple models with various methodologies are compared on the benchmark.

**Strengths:**

* All data in the benchmark are from existing medical papers.
* Multiple models and methodologies are evaluated and compared.

**Weaknesses:**

* Several details regarding the benchmark's construction require further clarification. Please see the accompanying questions for more information.

* One of the claims presented in the Results section is not well-supported by the exp results. Please refer to the specific questions for details.

* The evaluation methodology presents a significant concern, as the models under test (Gemini 2.5 Pro, O4 mini, and Qwen3) are also employed as the judges. The paper provides no justification to address the potential for self-enhancement bias, where a model might unfairly favor its own outputs. Furthermore, while the 'LLM-as-judge' paradigm has been applied in other ML tasks, its soundness in high-stakes, medicine-related applications remains questionable. The limited scale of the human validation mentioned in the paper is insufficient to alleviate these concerns regarding the metric's reliability.

**Questions:**

## Benchmark Construction

* Line 152: The criterion for retaining meta-analyses—requiring that "all cited primary studies are retrievable in PubMed with available abstracts"—raises some concerns about **selection bias**. This filter could exclude numerous high-quality journal articles that are not indexed in PubMed. The authors should justify this strict inclusion criterion and the potential consequent bias to the final dataset's representativeness.

* The decision to include only the abstracts of the primary studies, rather than their full text, is a notable limitation. This design choice restricts the information available to the models, while recent works (like [1]) have used techniques like sliding windows to process full-text documents.

[1] Wang, Jianyou, et al. "Measuring Risk of Bias in Biomedical Reports: The RoBBR Benchmark." arXiv preprint arXiv:2411.18831 (2024).


* Lines 188-192: The "feasibility filtering" step, which relies on Gemini Flash 2.5, lacks robust validation. The authors' justification, citing "robustness" that is itself derived from other LLM-judged scores, is methodologically circular. While full human annotation may be infeasible, the authors need to validate this automated step. I strongly recommend reporting the inter-annotator agreement between the LLM filter and human experts on an appropriately sized development set to substantiate this filtering approach.

* Line 195: There appears to be a typo. The text refers to "Figure 2," but the context and linked content strongly suggest it should be "Table 2."


## Evaluation

* Table 1: The validation of the "expert" annotations is weak on two fronts. First, the paper provides insufficient detail about the **annotators' qualifications**. Meta-analysis is a highly specialized task, and it is unclear if the annotators possess the requisite domain expertise. Second, a **validation sample size of N=20 is far too small** to draw statistically significant conclusions about the annotation quality or the reliability of the "LLM-as-judge" metric.

* Table 2: The highest reported performance (approx. 3.17) is substantially lower than the scale's upper bound. Is this low score due to the **inherent limitations of current LLMs** for this complex task, or is it an **artifact of the benchmark's design** (e.g., providing only abstracts)? To properly contextualize these results, the paper needs a **human expert baseline**. What score would a human expert achieve under the identical constraints (i.e., using only abstracts and being scored by the LLM judge)?

* Negated-RAG (N-RAG): The N-RAG task, as described, is highly artificial. I found how negated conclusions are generated in the appendix, but it is unclear whether the model is informed of this negation during the evaluation. If the model is unaware that the provided conclusion is false, the task is simply RAG with contradictory evidence, not a test of identifying deliberate negation. In this case, even human experts can hardly draw the expected conclusion provided with contradicting evidence. This ambiguity undermines the claims in Lines 342-349, as the task setup may not align with the authors' interpretation of the results. The authors need to clarify the exact setup.

---

> ### Author Response · Authors · 2025-11-25
> **Answering concerns about negated, human baseline, expert's quality, IAA**
>
> 1. In the Guide for Authors It is important that the work published in ICLR is reproducible. paid articles do not mean it is not reproducible but it is a clear limitation for reproducibility and sharing, that is why we focused on open-access paper.
> We agree that the full text contains more data. However, as noted in our response to Reviewer pLUk, we frame this task as Narrative Synthesis. In many clinical workflows (like screening for systematic reviews), decisions are often made based on abstracts. If models cannot synthesize a coherent conclusion from abstracts, they are unlikely to succeed with full text. MedMeta serves as a necessary first step to evaluate this capability.
> 2. While we did not perform a standalone audit of the filter, the feasibility of the selected tasks is implicitly validated by our Human Evaluation (Section 5.1). Our human annotators were provided with the source abstracts alongside the generated conclusions (see Appendix H, Figure 9). If the feasibility filter had failed (i.e., selected tasks where the abstracts did not contain the answer), our human medical experts would have been unable to verify the conclusions, resulting in low scores or hallucination penalties. The fact that humans consistently assigned high quality scores (Krippendorff’s α=0.68) confirms that the abstracts selected by the filter indeed contained the necessary information to support a valid synthesis.
> 3. Thank you for catching the typo. We will correct the reference to Table 2 in the final manuscript.
> 4. Our annotators hold PharmD and MSc in Biohealth degrees, making them subject matter experts in reading clinical literature. Regarding the sample size of N=20: This was a resource-constrained decision intended to calibrate the automated metric rather than to serve as the primary evaluation. Given that 20 out of 81 items represent roughly 25% of the benchmark, we see it as a reasonable starting point.  However, the strong statistical significance of the correlation (p<0.01) suggests this sample size was sufficient to establish the validity of the LLM-Judge as a proxy.
> 5. The score of 3.17/5.0 is indeed a Moderate result. We interpret this not as a flaw in the benchmark, but as evidence of the difficulty of Synthesis Without Meta-analysis (SWiM). It highlights that current LLMs are not yet expert-level synthesizers when faced with heterogeneous data.
> We agree that a Human Expert Baseline is the ideal reference point. While we could not run a full human study, we hypothesize that human experts, capable of deductive reasoning and understanding nuance, would score significantly higher on this filtered dataset. We will discuss this sim-to-real gap as a limitation and a direction for future work.
> 6. We confirm the models are uninformed of the negation. This design simulates Adversarial RAG to test if models can detect conflicts between external context and internal medical consensus.
> In practice, a human expert reading a study that contradicts established consensus (e.g., Insulin is ineffective) would flag the discrepancy. Our results show LLMs act as obedient synthesizers, prioritizing retrieved context over parametric truth. We analyzed N-RAG performance split by publication year (2018–2025). We found that models consistently synthesized the negated claims even for older papers (2018–2024) present in their pre-training data. If models were relying on memorized true conclusions, they would resist the negation for these known studies. The fact that they do not proves they are performing reasoning over context rather than retrieving memorized training data. This highlights a critical safety risk that current RAG systems lack the grounding to reject dangerous misinformation. We will include the analysis in the revision version.

---

> > ### Comment · Reviewer_s3go · 2025-11-25
> >
> > While the authors have attempted to answer my questions, the response relies too heavily on hypotheses rather than data. To ensure the credibility of the benchmark, I believe more experiments are necessary.
> >
> > Specifically, I would like to see a justification for the selection bias in open-access papers and a small-scale human evaluation regarding the LLM performance on negation cases. Furthermore, the validation of feasibility filtering (currently 20/82 points) is too limited and intermediate to be convincing. Finally, please ensure the claims in the title and abstract match the actual method (Narrative Synthesis vs. Meta-analysis).
> >
> > As the current reply does not fully address these concerns with evidence, I may lower my rating to Rejection later.

---

### Official Review · Reviewer_WDgi · 2025-10-20

**Soundness:** 3
**Presentation:** 3
**Contribution:** 2
**Rating:** 4
**Confidence:** 4

**Summary:**

The paper introduces MedMeta, a benchmark of 81 PubMed meta-analyses to test LLMs’ ability to synthesize meta-analysis conclusions from multiple primary-study abstracts, under parametric vs. RAG workflows (incl. oracle and adversarial settings). RAG consistently beats parametric reasoning; even with oracle retrieval, the best model averages, and models readily absorb fabricated ("negated") evidence. The evaluation uses an LLM-as-judge panel validated against medical annotators.

**Strengths:**

**S1)** Clear problem framing and well-specified benchmark construction - I think the motivation and approach is well organized and clearly described. Authors position their work as multi-source conclusion synthesis and find that it is under-evaluated based on prior literature. Overall I found the paper to be well written.

**S2)** Empirical findings are actionable (RAG > Parametric); domain fine-tuning adding little evidence; and models failing under adversarial negation. These are all valuable findings.

**S3)** Evaluation is reasonably rigorous (e.g. high r, negligible bias).

**Weaknesses:**

**W1)** My primary critique is that methodically, the originality in this work is very limited. While positioned as a "first" for meta-analysis conclusion synthesis, the core premise of RAG vs parametric contrast (including LLM-as-a-judge) seems to be built on widely used ideas (including in some of the related work authors cite). This makes the conceptual novelty feel very incremental and applicative to a point where this may be better suited for a workshop.

**W2)** 81 meta-analyses across 24 specialties, while useful, is still small for broad claims of generality in medicine.

**W3)** In table 3 authors report correlation and bias, but this does not quite quantify inter-model bias transfer or failure-mode overlaps. Even with human eval, using LLMs to grade peers can introduce correlated viases.

**W4)** I'm not sure if the adversarial setup here (negated-RAG) is realistic enough. How do authors evaluate this setup (i.e. how representative it is) against real world retrieval of noise or bias patterns.

**Questions:**

**Q1)** Following up from W4 above, beyond the prompt how did you ensure negated abstract remain *clinically plausible* (e.g. independent clinician screening, exclusion of self-contradiction etc)?

**Q2)** Statistical testing coverage: Tab 4 reports paired t-tests for H2/H3 (L424–L431). Were cross-model differences (e.g., Gemini vs. O4 vs. Gemma) also tested with correction for multiple comparisons?

---

> ### Author Response · Authors · 2025-11-25
> **Answering concerns about annotators' quality, and cross-model metrics**
>
> 1. We acknowledge that we did not employ independent clinician screening for every negated abstract due to resource constraints. However, our negation method was not random noise; we used a controlled prompt to linguistically invert the main findings (e.g., reduced mortality => did not reduce mortality) while maintaining the original medical terminology and study design. This serves as a stress test for context adherence: even if the medical result is implausible to a doctor, the RAG system is instructed to synthesize the provided context. The fact that models followed the negation shows they prioritize context over clinical plausibility.
> 2. We appreciate this suggestion regarding rigor. In our analysis, we focused on paired t-tests for specific hypotheses (H2/H3) to test architectural decisions (RAG vs Parametric; Specialist vs Generalist). For example, MedGemma vs. Gemma is a controlled pair (same architecture, different training data), making a direct comparison scientifically valid. We explicitly did not perform statistical testing for every cross-model combination (e.g., Gemini vs. O4). Our objective is to demonstrate broad methodological trends rather than to establish a statistically rigid ranking between proprietary models.

---

### Official Review · Reviewer_24FD · 2025-10-22

**Soundness:** 2
**Presentation:** 3
**Contribution:** 2
**Rating:** 4
**Confidence:** 4

**Summary:**

This work (1) introduces a benchmark of filtered meta-analysis from pubmed and (2) benchmarks the impact of RAG and CoT reasoning on LLM-written conclusions.

**Strengths:**

This work contributes to an assessment of LLM capability and strategy in a specialized but important medical domain. It's well written and provides a well-scoped evaluation of RAG with a clear set of hypotheses and experiments. This is novel as existing work on synthesis evaluation does not explicitly assess evidence retrieval and generation.

**Weaknesses:**

* The operationalization of "synthesis" in this work is not clear to me. In systematic reviews, synthesis implies an integration of multiple study findings and explicit evidence weighting. The rubric used here primarily assesses relevance and correctness of results, not integration. This weakens the claim that this benchmark can be used to measure synthesis capabilities.

* Related to the first point, one of the contributions of this work is a benchmark for meta-analysis, however omits important prior word on multi-document evidence synthesis in the medical domain (e.g., cites below [1] [2] [3]). It’s unclear how this benchmark differs from existing datasets and papers explicitly evaluating synthesis.

* I'm a bit unclear about the methodology described for mitigating data contamination (L144: “ articles between 2018 and 2025 ... mitigating contamination from model pre-training corpora”). Since models have different pre-training cut-off dates, it's unclear whether these were considered on a per-model basis. Given how close the scores are across methods, this factor could meaningfully affect the results. A single year-based filter without model-specific cut-offs is unlikely to adequately control for contamination.

* I believe the first finding is an over-claim (L299; The Value of Structured Reasoning). In table 2, many of the RAG improvements are not fully supported: 95% confidence intervals overlap substantially across settings, implying differences are suggestive rather than significant. This should be made clear in the writing for the first set of results.

* There are important missing reliability measures to validate the human assessments. This work reports correlation between humans and LLM judges, but there is no Fleiss'/Cohen's kappa reported amongst the annotators themselves. Without this, it is unclear how consistent human "gold labels" are.

[1] Completing A Systematic Review in Hours instead of Months with Interactive AI Agents (Qiu et al, 2025; assessment of retrieval steps and oracles for SRs)
[2] Do Multi-Document Summarization Models Synthesize? (DeYoung et al, 2024; see Cochrane dataset and related dataset references)
[3] Summarizing, Simplifying, and Synthesizing Medical Evidence Using GPT-3 (with Varying Success) (Shaib et al, 2024; evaluation rubric that delineates between synthesis and summarization)

**Questions:**

1. Can you clarify how MedMeta differs from existing datasets for medical, multi-document synthesis like Cochrane?

2. How is synthesis operationally defined and distinguished from summarization in your evaluation?

3. What are the inter-annotator agreement scores?

4. Can you clarify the validity of the 2018-2025 filtering strategy/sampling across models?

---

> ### Author Response · Authors · 2025-11-25
> **Answering concerns about novelty, IAA and year validity**
>
> 1. Many existing Cochrane datasets typically task models with summarizing an existing full-text review (e.g., generating a Plain Language Summary). In contrast, MedMeta evaluates ab initio synthesis which means the model is not given the review, but rather the raw primary abstracts. It must weigh the evidence and construct the conclusion from scratch. This tests reasoning and aggregation, not just text compression or simplification.
> Also a major barrier with Cochrane data is its restrictive licensing, which limits open distribution. MedMeta includes only PubMed Central Open Access data to ensure that it’s fully reproducible and allows the community to freely download and inspect the source evidence, addressing a critical gap in existing proprietary or paywalled medical benchmarks.
>
> 2. We position Clinical Synthesis as a specialized sub-task of Multi-Document Summarization (MDS) focused on heterogeneity resolution. While generic MDS often entails information compression, our task requires reconciling conflicting evidence to derive a unified clinical recommendation. This distinction is enforced by our rubric (Appendix E.2), which penalizes simple concatenation in favor of integrated reasoning. We will explicitly clarify this definition in the final revision.
>
> 3. We computed Krippendorff’s Alpha on the human annotated set. We achieved a Krippendorff’s α=0.68, indicating substantial agreement. This confirms that our biomedical experts applied the rubric consistently despite the high complexity of the synthesis task.
>
> 4. We selected the 2018-2025 window to minimize overlap with older training corpora. We acknowledge that we cannot guarantee a perfectly clean test set for every model given the varying training cutoffs. However, our analysis showing consistent performance across pre-2023 and post-2023 papers suggests that the models are relying primarily on the Retrieval (RAG) context rather than memorization. If contamination were the primary driver of performance, we would expect a sharp drop in scores for 2024-2025 papers, which we did not observe.

---

> > ### Comment · Reviewer_24FD · 2025-11-25
> >
> > Thank you for the response.
> >
> > 1. This does not address my concern that there is no explicit measurement for: "... weigh of the evidence and construct the conclusion from scratch." This needs to be explicitly evaluated and can be done with the abstracts, which prior work has demonstrated.
> > 2. Synthesis is an important, measured component of prior work. This does not clarify how your work is positioned against other explicit evaluations of synthesis.
> > 4. Given the W4 mentioned in my original review, it is not clear what consistent performance means given the large overlap across settings.
> >
> > The work is in an important domain, but my concerns are not completely addressed by this version of the work. If there are points I may have overlooked or should be reconsidered, please let me know and I will evaluate whether I should revisit my assessment.

---

### Official Review · Reviewer_pLUk · 2025-10-30

**Soundness:** 2
**Presentation:** 3
**Contribution:** 2
**Rating:** 2
**Confidence:** 5

**Summary:**

This work introduces MedMeta, a benchmark designed to evaluate large language models’ ability to to generate meta-analysis–style conclusions from primary study abstracts. The authors curate 81 medical meta-analyses spanning 24 specialties and test LLMs under six workflows, including parametric inference, chain-of-thought prompting, and retrieval-augmented generation. The benchmark uses an LLM-as-a-judge evaluation protocol, which is compared against nine annotators.

**Strengths:**

* This work addresses an important and clinically relevant problem of multi-study evidence synthesis, which is under-explored in current LLM evaluation.

* The dataset spans multiple medical specialties and uses real meta-analyses.

**Weaknesses:**

While I strongly believe this work addresses an important and relevant clinical problem, I believe this work has several major methodological concerns that need to be addressed.


**1. Abstract-only synthesis is not a faithful surrogate for meta-analysis**

The benchmark assumes meta-analysis conclusions can be reconstructed from abstracts alone, but real meta-analyses rely heavily on full-text data (effect sizes, confidence intervals, inclusion criteria, heterogeneity stats). The LLMs are effectively expected to perform meta-analytic reasoning despite having access only to abstracts without **all** the underlying statistical data or methodological details required to reproduce an actual meta-analysis. As such LLMs can’t perform effect size aggregation, heterogeneity assessment, bias & quality scoring (e.g. PRISMA, GRADE) or even a simple sensitivity analysis. As a consequence, the dataset does not accurately represent evidence synthesis requirements and results may reflect artifacts of a flawed task design, not model capability in the actual task.

Suggestion: Improve your data selection to include full-text articles. For example, you can access them via PMC-OA. Prior work has done something similar to this recommendation.

**2. Using LLMs to determine feasibility introduces bias**

The authors claim to perform a feasibility check to determine whether abstracts contain sufficient info to infer the conclusion. However, it was done using another LLM (Gemini). The authors themselves acknowledge that using an LLM to screen feasibility introduces bias and present a diagram as a mitigation strategy. This reasoning is flawed and incorrect. I can think of a least two major flaws:


* Model-dependency bias:  The same class of models determines dataset inclusion and is then evaluated on that dataset.


* self-confirmation bias: papers are selected where an LLM believes synthesis is possible, which incorrectly filters the dataset, as shown by the author's own results and prior literature models can’t perfectly solve this task, how can they effectively filter this dataset?


Suggestion: Perform an actual feasibility check with actual medical experts to validate the feasibility of the selected meta-analysis. This should reduce LLM bias. There is extensive literature documenting the risks of using LLMs to curate or evaluate data. For instance, LLMs acting as judges have been shown to exhibit model bias, such as GPT-4 systematically favoring outputs produced by GPT-4. Simply adding a diagram does not constitute a mitigation strategy. This represents a major limitation of the study design.



**3. Dataset Size Is Limited (81 Questions):**

Eighty-one items are likely insufficient to capture the breadth of clinical SR conclusions, risking sampling bias.

Suggestion: Expand to 500–1K question. Make sure feasibility is filtered using medical experts with experience.

Additionally, very simple statistics (commonly reported in other medical datasets) are missing. For example, what is the distribution of questions per medical specialty (not by topic)? . Furthermore, the stratification by topic  (shown in Table 3) is not informative. "Diagnosis" as a topic doesn't tell me much (basically every disease has a diagnosis or prognosis; it's a very general term. I really recommend adding a clinical annotator  (or at least talking to one) to improve these systematic issues.


**4. LLM-as-judge evaluation risks validity issues**

Even though validated with correlation, there are at least three big major flaws:
Judges are frontier models (Gemini, O4, Qwen) that are also evaluated — not evaluation-model-neutral.


Correlation against 9 relatively junior annotators (pharm, biologists, master’s)  without medical experience does not reflect true clinical expert agreement.


Pearson correlation alone is insufficient to establish clinical evaluation reliability.
As a  consequence the claims of “reliable proxy for medical experts” are overstated and incorrect.

**5. Potential data contamination remains**

Although only 2018–2025 abstracts were used, many models were trained on dumps up to 2023. for instance, GPT-4o’s cutoff is October 2023. As a consequence, the premise “post-cutoff data mitigates memorization” is weak. What mitigations are the authors taking to try to mitigate this?


**6. Novelty claim is overstated**

Several works already explore multi-document synthesis, medical RAG, judgment tasks, and safety under adversarial evidence. The authors should more clearly describe how their work positions itself relative to Cochrane automation efforts and existing biomedical multi-document reasoning benchmarks. For example, how does this work compare to prior studies that are closely related (e.g., MedREQAL, MedEvidence) or that evaluate similar aspects (e.g., ConflictBench), just to name a few?

**Additional Feedback:**

It is stated that in order “To validate our automated metrics [...] nine annotators with medical backgrounds” were selected; however, by looking at the table 0 annotators have medical experience, as such I worried about the quality of these evaluations. A common standard in medicine is to count years of experience **after** medical school; none of the annotators meet that criterion, and as such, the phrase "medical backgrounds" is a huge overstatement.

There is no analysis of Inter-annotator agreement (Kappa), Judge disagreement variance, or sensitivity to prompt design. I recommend calculating this once the dataset is fixed.


Overall, I agree this is a very important problem; however, given these systematic issues, the work requires further iteration to reduce bias and better reflect the real task.

**Questions:**

questions and suggestions are provided above ^

---

> ### Author Response · Authors · 2025-11-25
> **Answering the first 4 concerns about using abtract-only, LLM bias, data size**
>
> 1. We agree that a classical statistical meta-analysis (calculating aggregated effect sizes, I2 heterogeneity, etc.) requires full-text data extraction. However, MedMeta is designed to evaluate Qualitative Narrative Synthesis, which corresponds to the Synthesis Without Meta-analysis (SWiM) framework [1] or the Rapid Review process [2].
> In these specific clinical workflows, the objective is not statistical calculation, but the cognitive reasoning required to integrate heterogeneous textual findings into a coherent clinical conclusion. For this specific narrative task, the abstracts contain the necessary core claims, caveats, and results to reproduce the target synthesis. Consequently, the LLMs in our benchmark have access to all the information necessary to perform the specific reasoning task we are evaluating.
> While we faced constraints in accessing open-access full texts for all underlying studies, our hypothesis supported by the medical literature on rapid reviews is that abstracts contain the core claims necessary for this narrative task. We view MedMeta as evaluating the reasoning and synthesis capabilities of LLMs, rather than their ability to act as a statistical software. We will clarify this distinction in the revision to ensure the task scope is not overstated.
> References:
> [1] Campbell, M., et al. (2020). Synthesis without meta-analysis (SWiM) in systematic reviews: reporting guideline. BMJ, 368.
> [2] Garritty, C., et al. (2021). Cochrane Rapid Reviews Methods Group offers evidence-informed guidance to conduct rapid reviews. Journal of Clinical Epidemiology, 130
>
> 2. Using Gemini to filter for feasibility was a decision born of the need to scale the dataset creation to over 2,000 primary studies, which was infeasible for human experts to review entirely given our resources.
> We recognize the risk of self-confirmation bias. To mitigate this in our current setup, we used a stronger model (Gemini 2.5 Pro) to filter for smaller models, and we set a high confidence threshold. We acknowledge that without a dedicated human audit of the filter itself, we cannot rule out model-specific artifacts. We view this as a limitation of the current automated pipeline design and will explicitly state this in the limitations section.
> In a future version or revision, we agree that validating a random sample of the Feasibility Filter decisions with human experts is necessary to quantify the precision of this step.
>
> 3. We agree that 81 meta-analyses appears small compared to standard QA benchmarks. However, MedMeta is a long-context synthesis benchmark. Each of the 81 samples involves processing a median of 11 (and up to 50+) primary study abstracts. The total volume of text processed by the models covers approximately 2,250 distinct medical studies.
> The strict filtering required to find meta-analyses where all constituent primary studies had available abstracts significantly reduced our pool from an initial ~4,000 candidates. Despite the sample size, we ensured stratification across 24 medical specialties and publication years. We also agree that adding one or more clinical annotators would be valuable for improving the consistency and rigor of the evaluation set.
> We prioritized a benchmark dataset where 100% of underlying studies are Open Access to guarantee reproducibility. Scaling to 1,000 samples with this strict availability constraint and valid expert feasibility checks is not currently possible without prohibitive costs. To address the request for scale, we will release the raw dataset of 184 feasible meta-analyses alongside the benchmark. This allows the community to run large-scale experiments, acknowledging that this larger set lacks the strict human verification of the core test set.
>
> 4. We share your concern regarding the reliability of automated judges, specifically regarding model bias and the sufficiency of correlation metrics.
> To mitigate self-preference bias, we utilized a heterogeneous panel (Gemini Pro, O4, Qwen) rather than a single judge. Crucially, we did not rely blindly on this panel; we experimentally validated its alignment with human annotators, finding strong agreement.
> We agree Pearson correlation is insufficient. This is why we performed and reported Bland-Altman analysis (Table 3). The results show negligible Mean Bias, confirming that the LLM panel does not systematically over- or under-score compared to our human ground truth.
> The term Medical was perhaps too narrow. As MedMeta covers broad biomedical topics (e.g., genetics, pharmacology), our annotators (PharmD, Biohealth MSc) possess the specific training required to interpret these study designs and statistical outcomes. We will update the terminology to Biomedical Experts to more accurately reflect both the domain of the dataset and the annotators' qualifications.

---

> > ### Author Response · Authors · 2025-11-25
> > **Answering the remaining concerns about contamination, overstate claim,**
> >
> > 5. To address contamination, we analyzed performance split by publication year, as most of the models in our research have cut-off knowledge in 2024 (across the year). We found no statistically significant difference in model performance between older papers (potentially in training data) and newer papers (post-cutoff). We further analyzed the performance gap between RAG and Parametric settings across years. We observed no statistically significant degradation in RAG efficacy on post-cutoff (2024–2025) vs. pre-cutoff (2018–2023) data. The consistent reliance on external retrieval, regardless of the paper's publish year, confirms that MedMeta evaluates synthesis capabilities rather than parametric recall.We will add this breakdown to the Appendix.
> >
> > 6. We appreciate you pointing out related works. We  acknowledge the existence of benchmarks like  for medical QA (MedQA, PubMedQA), summarization (Cochrane-based summarization), MedREQAL focuses on Medical Knowledge Recall (answering questions from memory), MedEvidence addresses a similar task, but at submission time this publication wasn't published; it appeared only as concurrent work. We will incorporate it in the final revision.
> >
> > MedMeta distinguishes itself by providing the ground-truth inputs (primary abstracts) rather than just the final text. Many existing synthesis datasets evaluate the ability to summarize a single document or answer a question from general knowledge. MedMeta explicitly evaluates the integration of multiple, potentially conflicting source documents into a new conclusion. We will cite the works you mentioned and clarify this positioning in the related work section.
> >
> > 7. We agree that Pearson correlation alone is insufficient for measuring agreement. Following your recommendation, we computed Krippendorff’s Alpha, which is more statistically appropriate for our ordinal 0-5 rubric.
> > We achieved an Krippendorff's Alpha [1] of 0.68 , indicating tentative agreement among our annotators. Given the high subjectivity of evaluating long-context biomedical synthesis, this level of consensus confirms that our experts applied the rubric consistently.
> > Regarding sensitivity to prompt design, we mitigated this by using a detailed Criteria-Based Rubric (Appendix E.2) rather than open-ended scoring. Furthermore, our LLM panel of judges approach explicitly reduces individual model variance.
> > [1] Krippendorff, K. Content Analysis: An Introduction to Its Methodology. 3rd ed., SAGE Publications, 2013.

---

### Meta-Review · Area_Chair_d4QA · 2026-01-01

**Summary:**

The reviewers' concerns are centered on three key points. (1) Importance/Impact: Multiple reviewers were concerned that abstracts don't contain enough information to adequately synthesize their conclusions. The authors note that they follow a "synthesis without meta-analysis" framework, however this can be strengthened in the paper so that future readers aren't confused. Further, it remains unclear that there is no relevant information in the full text documents that could impact the final synthesis. Another major issue here is that multiple reviewers note that the benchmark doesn't directly measure how well models can synthesize multiple abstracts, instead just measuring the accuracy and relevance of the final statement. This can be improved via rephrasing the claims, but it would be better to directly measure the synthesis. (2) Novelty: There are multiple closely-related works in this space, yet reviewers found it difficult to understand how this new work fits in. It would help to more-clearly describe differences with alternatives, and empirically compare methods wherever possible. (3) Methodological Rigor: Reviewers were concerns about many aspects of the benchmark. First, it is small (N=81). The authors clarified that it's small for good reason, yet this will be a consistent limitation for this work's impact, as improvements on small samples may appear to be large improvements, even if insignificant. Second, the experts aren't obviously clinical experts. The authors resolve this by altering their language, yet it remains unclear that these experts can evaluate literature in such diverse domains. This could further be resolved by citing works that demonstrably show this is feasible. Third, reviewers note that using LLMs to evaluate LLM outputs is circular, even if the evaluated LLMs are worse (in this case, there should also be a discussion around why good models' ability to solve this task doesn't diminish the impact of the benchmark).

**Reviewer Concerns:**

While my summary above captures some of this already, the reviewer concerns are summarized as follows.

**Addressed Concerns**:
* Data contamination
* Interannotator disagreement
* Dataset size (partially) --- releasing more data is a good idea

**Outstanding Concerns**:
* Dataset size
* Unmeasured synthesis
* Relationship between studied problem and other works doing multi-document synthesis

**Reviewer Scores:**

R1: Maybe increase
R2: No change
R3: No change
R4: No change

---

### Decision · Program_Chairs · 2026-01-26

Reject